# In situ observation of coalescence of nuclei in colloidal crystal-crystal transitions

Yi Peng [1,2,3] ✉, Wei Li [2], Tim Still[4], Arjun G. Yodh [4] & Yilong Han [2] ✉

Coalescence of nuclei in phase transitions significantly influences the transition rate and the properties of product materials, but these processes occur rapidly and are difficult to observe at the microscopic scale. Here, we directly image the coalescence of nuclei with single particle resolution during the crystal-crystal transition from a multilayer square to triangular lattices. The coalescence process exhibits three similar stages across a variety of scenarios: coupled growth of two nuclei, their attachment, and relaxation of the coalesced nucleus. The kinetics vary with nucleus size, interface, and lattice orientation; the kinetics include acceleration of nucleus growth, small nucleus liquefaction, and generation/annihilation of defects. Related mechanisms, such as strain induced by nucleus growth and the lower energy of liquid-crystal versus crystal-crystal interfaces, appear to be common to both atomic and colloidal crystals.

Crystal-crystal (c-c) transitions between different crystalline lattices occur widely, for example in metallurgy, in the earth's mantle, and in nanocrystal systems. Moreover, these solid-solid transitions have consequences for steel production and for the properties of memory alloys and man-made diamond[1–3]. A typical c-c transition involves four stages: (I) Incubation wherein subcritical nuclei form and disappear in the metastable parent crystals; (II) Formation of nuclei of critical size; (III) Growth of post-critical nuclei and their coalescence; (IV) Ripening stage of polycrystalline grains. All these stages influence the phase-transition rate, and the final structure and material properties. However, unlike stage IV which occurs on a mesoscopic scale, the first three stages have features at comparatively small spatial scales and occur rapidly on time scales that are difficult to observe.

C-c transitions are also more complex than the better studied crystallization and melting transitions. In c-c transitions, the parent and product lattices often lack group-subgroup symmetry, which can lead to complicated multistep kinetic pathways[4], and the nuclei can grow via random particle diffusion or collective motion (i.e., martensitic nucleation). Additionally, in c-c transitions, nucleus growth can distort the parent lattice which strongly influences the transition kinetics, and defects and crystalline interfaces with different structures and energy complicate c-c transitions. In this contribution we explore

coalescence of nuclei; this phenomenon has been rarely explored even in simulation, in part because observation requires a wider range of spatial and time scales, e.g., compared to single nucleus formation in c-c transitions[5–8].

Here we employ colloids to push beyond these prior limitations. Colloids are attractive model systems for the study of phase transitions. This is because the micron-sized particles can be directly visualized and their motions tracked inside the bulk using optical microscopy[9]. C-c transitions in colloidal systems can be induced by applying electric or magnetic fields[10,11], or by tuning particle size and interaction[12–18]. To date these studies have revealed a rich variety of kinetic processes such as martensitic transformations within small crystallites[15] and under external stress[13,19], two-step diffusive nucleation with intermediate liquid states[12], forward-martensitic-and-reverse-diffusive transitions[11], and softness-dependent transition pathways[14,20]. These results, discovered in colloidal model systems, have cast new light on related processes in atomic systems. For example, the two-step nucleation with an intermediate liquid state was subsequently observed in metals[21].

Notably, however, the previous studies on the nucleation and growth focus on the evolution of an individual nucleus (stages I and II)[10–18]. The coalescence of nuclei (in stage III) has not been elucidated.

[1]Beijing National Laboratory for Condensed Matter Physics, Institute of Physics, Chinese Academy of Sciences, Beijing 100190, China. [2]Department of Physics, Hong Kong University of Science and Technology, Hong Kong, China. [3]School of Physical Sciences, University of Chinese Academy of Sciences, Beijing 100049, China. [4]Department of Physics and Astronomy, University of Pennsylvania, Philadelphia, PA 19104, USA. ✉e-mail: pengy@iphy.ac.cn; yilong@ust.hk

To date, coalescence of nuclei has been studied in supersaturated solutions during crystallization by high-resolution transmission electron microscopy (TEM)[22–26], but such phenomena have not been investigated in c-c transitions because in situ observation inside bulk crystals with single-particle resolution is experimentally challenging.

In the present contribution, we visualise the coalescence of nuclei in situ with single-particle resolution in c-c transitions with different nucleus sizes, interface coherences, and lattice directions. Three types of kinetic pathways for homogeneous nucleation inside crystalline domains and two types for heterogeneous nucleation on grain boundaries are uncovered.

## Results

### Experiment

To drive a c-c transition, the colloidal crystal needs to be tunable. Here we employ monodisperse poly(N-isopropylacrylamide) (NIPA or pNI-PAM) microgel spheres whose diameter $\sigma$ linearly decreases from 0.76 $\mu$m at 26.4 °C to 0.67 $\mu$m at 30.6 °C (Supplementary Fig. 1a)[27]. The particles interact via short-range repulsion[28] and exhibit phase behaviours quite similar to those of hard spheres[27,29].

NIPA spheres confined between two plates self-assemble into a cascade of crystalline phases. As plate separation ($H$) increases, the colloidal system evolves in the following sequence: $1\triangle, 2\square, 2\triangle, 3\square, 3\triangle, \ldots$[30–32]. Here $1\triangle$ denotes a one-layer (monolayer) triangular lattice, $2\square$ denotes two-layer square lattice, etc. Similar phases have been observed in plasmas[33] and in electron bilayers within semiconductors[34]. The phase diagram of hard spheres as a function of particle volume fraction ($\phi$) and the reduced plate separation or sample height ($H/\sigma$) has been precisely investigated in simulations[30,31]. The volume fraction, or packing fraction, is the ratio of the total volume of spheres to the volume of the sample. By varying the particle diameter ($\sigma$) with temperature, we can experimentally tune both $\phi$ and $H/\sigma$, and we can generate $n\square \rightarrow (n-1)\triangle$ transitions[12,13]. We observe similar behaviours in $n\square \rightarrow (n-1)\triangle$ transitions ($n = 5, 6$) and illustrate them using $5\square \rightarrow 4\triangle$ as examples.

To ensure that coalescence of nuclei occurs in the chosen field of view, we heat the selected region constantly with a beam of light (Supplementary Fig. 2)[29]. The heated area is set at a temperature $T_{amb} + \delta T$; the ambient temperature $T_{amb}$ is controlled by an objective heater on a microscope with 0.1 °C resolution. $\delta T = 1.6$ °C is the local optical heating effect, which is reached within 3 s after the light is switched on (Supplementary Fig. 1b)[29].

The c-c transition occurs when $T_{amb} < T_{c-c} < T_{amb} + \delta T < T_m$ (i.e., $\phi_{amb} > \phi_{c-c} > \phi_{amb} + \delta\phi > \phi_m$), where $T_{c-c}$ and $T_m$ are the c-c transition and melting points, respectively. The temperature is uniform in the central $\pi(38\,\mu m)^2$ area (i.e., $10^5$ particles per layer) of the $xy$ plane (Supplementary Fig. 2b) and in the $z$ direction for such thin films[29]. The nucleation and growth of $\triangle$-lattice from a superheated metastable $\square$-lattice at a fixed $T$ (i.e., at fixed $\phi$ and $H/\sigma$) is observed with an optical microscope. Particle motions are recorded with a CCD (charge-coupled device) camera at 10 frames/s, and the particle trajectories are tracked from image analysis[35]. Experimental details are provided more fully in the Methods section.

### Nucleus coalescence inside crystalline domain

Stages I and II of the c-c transition in thin-film NIPA colloidal crystals have been studied in refs. 12,13. Under isotropic pressure, the transition exhibits two-step diffusive nucleation ($n\square \rightarrow$ liquid $\rightarrow (n-1)\triangle$)[12], but with a small pressure gradient the transition exhibits one-step martensitic nucleation at the early stage, followed by a diffusive growth[13]. Here, we observe the coalescence of two post-critical $\triangle$-lattice nuclei; this occurs under isotropic stress in stage III of the c-c transition. The observed processes can be classified into five types of kinetic pathways: (1) Nuclei with parallel lattices that coalesce into a single crystallite inside the parent lattice (Fig. 1, Supplementary Fig. 3);

(2) Nuclei with angled (not parallel) lattices that coalesce to a crystallite with a low-angle or high-angle grain boundary (GB) (Fig. 2, Supplementary Fig. 4); (3) A small nucleus that liquefies and is then attracted to a large nucleus nearby (Fig. 3); (4) Coalescence on a low-angle GB involving liquid surfaces (Fig. 4); (5) Coalescence on a high-angle GB (Fig. 5).

Type (1): Two nuclei have parallel $\triangle$ lattices and incoherent $\triangle$-$\square$ interfaces, as shown in Fig. 1 and Supplementary Mov. 1. Both nuclei have a special misorientation angle, $\beta_1 = 45°$, relative to the parent $\square$ lattices (Fig. 1b), i.e., 45° between the [01] directions of the nucleus and the parent lattice. This special orientation relation is a feature of their early-stage martensitic nucleation via collective particle motions[13]. The nuclei grow in a diffusive manner in the late stage, which does not change $\beta_1$ considerably[13]. The nuclei have incoherent surfaces, i.e., the two lattices do not match on the $\triangle$-$\square$ interface.

Before 1090 s, the centres of these nuclei barely move, and their diameters grow linearly, i.e., according to the Wilson-Frenkel law[36,37]. Thus, the approach speed of the two $\triangle$-$\square$ interfaces (labelled by the parallel yellow lines in Fig. 1b) is a constant. The approach speed doubles after $t = 1090$ s, i.e., when the separation between the two nuclei $d < 11a$ (Fig. 1g); here $a$ is the lattice constant. A region between the two nuclei labelled by the yellow rectangle in Fig. 1c is characterized by its lattice orientation angle ($\alpha$) with respect to the x-axis of the lab frame, and the magnitude of the four-fold orientational-order parameter, i.e., $|\psi_4|$. The orientation order parameter $\psi_m = \langle e^{im\theta_{jk}} \rangle$, where $\theta_{jk}$ is the angle between the bonds with the nearest neighbour particles $j$ and $k$, and $\langle \rangle$ represents an ensemble average. $\psi_m$ with $m = 4$, 6 represent four- and six-fold orientation order, which are for $\square$ and $\triangle$ lattices, respectively. $\alpha$ and $|\psi_4|$ are constant before 1090 s. The $\square$ lattice in the boxed region (labelled in Fig. 1c) rotates and becomes progressively more distorted when $d < 11a$ and $t > 1090$ s (Fig. 1b, c, h). Such rotation may generate small defects such as two interstitials shown in Fig. 1c which quickly transform to the triangular lattices. The $\triangle$ and $\square$ lattices have the same lattice constant; thus, $N$ spheres confined by two walls with separation $H$ occupy volume $\frac{NHa^2}{5}$ in $5\square$ crystal and volume $\frac{\sqrt{3}NHa^2}{8}$ in $4\triangle$ crystal. The relative volume change in a $5\square \rightarrow 4\triangle$ transition is 8%. The $\square$ lattice has a lower in-plane area density and becomes distorted as the $4\triangle$-lattice nuclei grow. When $d < 11a$, the boxed region in Fig. 1c becomes more compressed and disordered by both nuclei compared to other regions. When the closest surfaces of the two nuclei are separated by less than approximately $3a$, then a thin channel of triangular lattice develops and connects the nuclei (Fig. 1d, e); as a result, the coalesced nucleus acquires a concave shape (Fig. 1e). The concave part rapidly grows until it becomes convex to reduce the interfacial energy (Fig. 1e, f).

We also investigated situations wherein the two closest $\triangle$-$\square$ interfaces of parallel $\triangle$-lattice nuclei are coherent. In this case, the kinetics are similar to the those when the interfaces are incoherent except that the nuclei temporarily stop approaching each other before their coalescence (Supplementary Fig. 3e). The coherent interface propagates by a ledge mechanism with row-by-row growth (Supplementary Fig. 3b, c)[1], which compresses the $\square$ lattice along its [10] direction. Consequently, particles between the two nuclei become less mobile, and the interface migrations temporarily stop for 40 s (Supplementary Fig. 3b, e). The subsequent ledge growth of the two coherent interfaces rotates the parent lattice between them, thereby enhancing nucleus growth and facilitating coalescence.

Type (2): For two nuclei with angled (not parallel) lattices, the coalescence produces either a low-angle or a high-angle GB which subsequently anneals away, i.e., via two kinetic pathway types: (2a) (Fig. 2b–d) and (2b) (Fig. 2e–g). When the misorientation angle of the two nuclei is $\beta_2 \lesssim 10°$, then the interface of two fused nuclei is a low-angle GB, which can be viewed as a chain of dislocations whose Burgers vectors are parallel (Fig. 2b). These dislocations move under thermal fluctuations and elastic stresses in the lattice[38] and are attracted by the

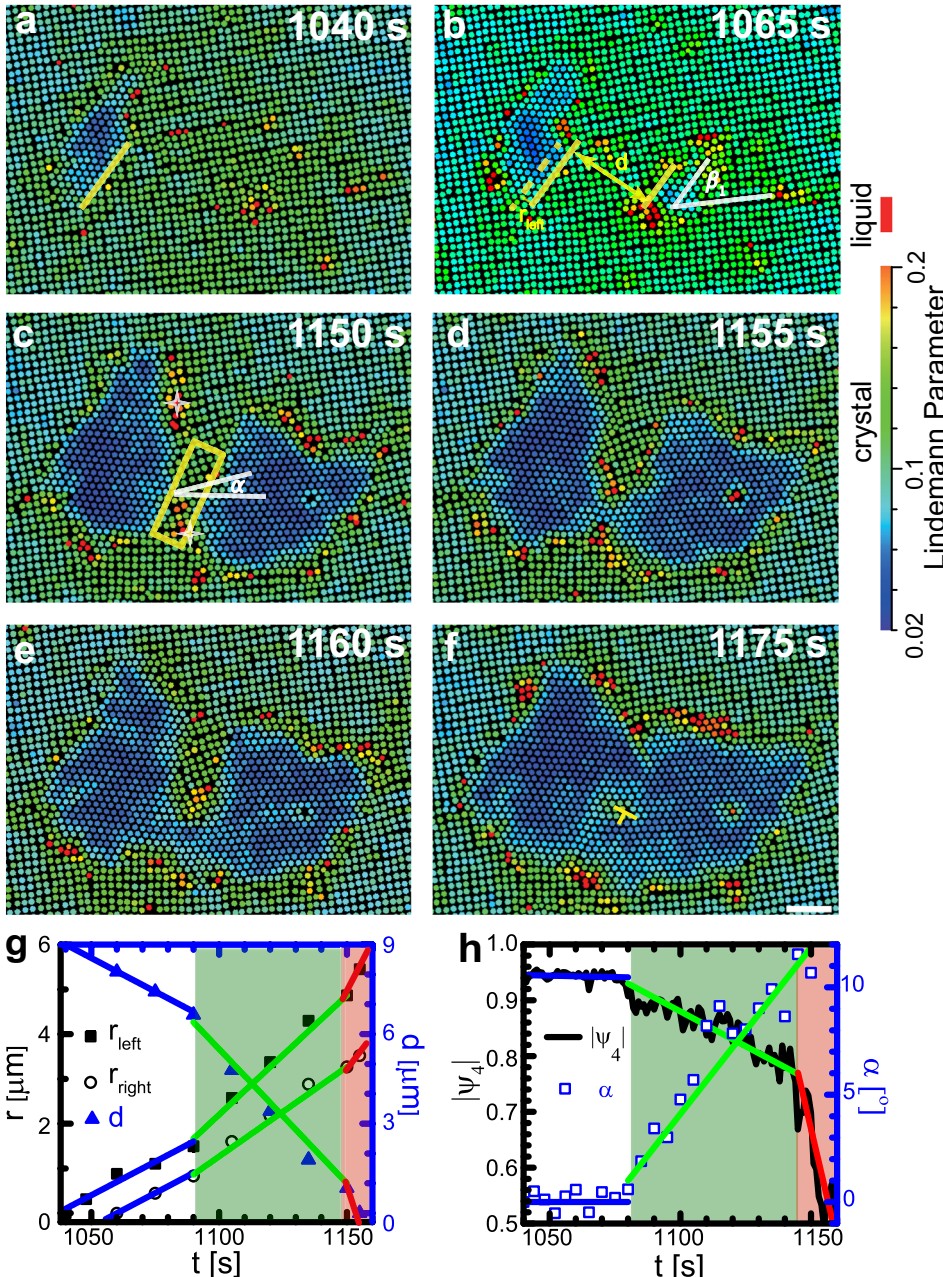

**Fig. 1 | Coalescence of two nuclei with parallel triangle lattices and incoherent triangle-square interfaces.** The colours in (**a**–**f**) represent the dynamic Lindemann parameter for each particle measured within 4 s (Methods). Heating light is switched on at $t = 0$. Scale: 5 $\mu$m. **a** At 1040 s, one nucleus forms. **b** Another nucleus forms at 1060 s. The yellow solid lines indicate the closest facets of the two nuclei. The misorientation angle is $\beta_1 = 45°$, between the [10] directions of the square and triangle lattices. The dashed line shows the interface position of the left nucleus at 1040 s, and $r_{left}$ measures the distance that the interface moves. **c** The growth of the two nuclei distort the square lattice between them. $\alpha$ is defined as the angle between the [10] direction of square-lattice and x-axis of the lab frame. The white hollow stars mark two interstitials generated by the lattice distortion. **d**, **e** The two nuclei coalesce at 1155 s by forming a thin channel of triangle lattice. **f** The fully coalesced nucleus contains a dislocation (⊥) owing to the slight mismatch of the orientations of the two triangle lattices. **g** The displacements $r$ of the two nuclei's facets with respect to their initial positions (yellow lines in (**a**, **b**)) and their separation $d$. The green and red regions represent the stages that nuclei grow faster and merge together, respectively. **h** The evolution of the four-fold orientational order parameter, $|\psi_4|$, and the [01] direction $\alpha$ of the square-lattice in the yellow rectangle region in (**c**). The blue, green and red lines in (**g**) and (**h**) indicate the evolution trend of parameters in various stages.

nearby GBs (Fig. 2c, d), resulting in disintegration of the low-angle GB. Dislocations randomly diffuse and eventually are either absorbed by the nucleus surface or stay inside the crystal domains (Fig. 2c, d). When $\beta_2 \gtrsim 10°$, then the coalescence generates a high-angle GB, which sweeps through the small nucleus and is eventually absorbed by the nucleus surface (Fig. 2e–g, Supplementary Fig. 4). The small nucleus slightly rotates and reduces $\beta_2$ from 23° to 15° when the GB sweeps through it (Fig. 2e–g). Note, in crystallization, atomic crystalline nuclei in liquid

solutions are observed to rotate and perfectly align their lattice orientations just before fusion, resulting in a single-crystal nucleus without a GB[22,24,25] or a twin structure[26]. This phenomenon is referred to as oriented attachment. By contrast, nuclei embedded in a solid (e.g., Fig. 2a and Supplementary Fig. 4) cannot easily rotate to align their lattice orientation before they fuse.

Type (3): When one nucleus is small with a large $\beta_2$, it liquefies as it approaches the large nucleus especially at low $\phi$. In Fig. 3 and

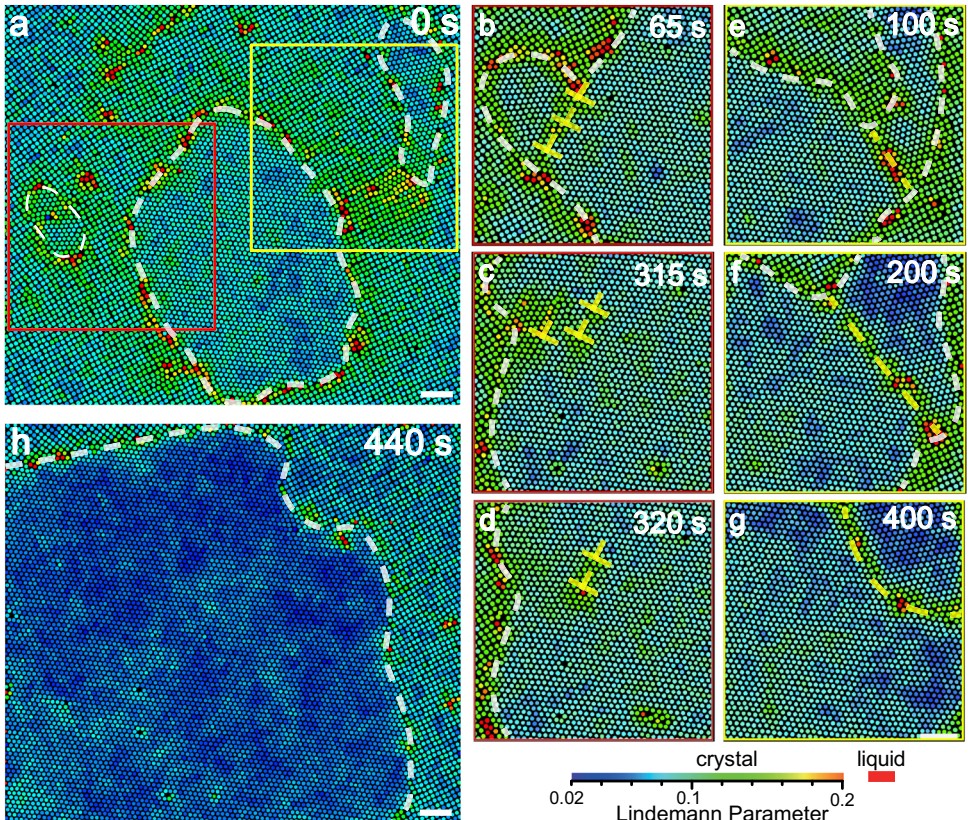

**Fig. 2 | Coalescence of nuclei with angled (not parallel) lattices. a** Three triangle-lattice nuclei embedded inside a square lattice. The left nucleus in the red box and the right one in the yellow box merge with the middle large nucleus. **b–d** Nuclei come in contact to form a low-angle GB in the region labeled by the red box in (**a**). The three dislocations (yellow ⊥) on the low-angle GB diffuse independently. One is absorbed by the nearby square-triangle interface, and the other two diffuse inside the triangle lattice. **e–g** Nuclei coalesce to develop a high-angle GB in the region labeled by the yellow box in (**a**). The high-angle GB (yellow dashed line) propagates through the smaller nucleus. **h** The three nuclei completely coalesce. Given that the nucleus at 440 s is out of the initial field of view in (**a**), the field of view in (**h**) slightly shifts upright relative to (**a**). The colours represent the dynamic Lindemann parameter for each particle. Scale bar: 5 μm.

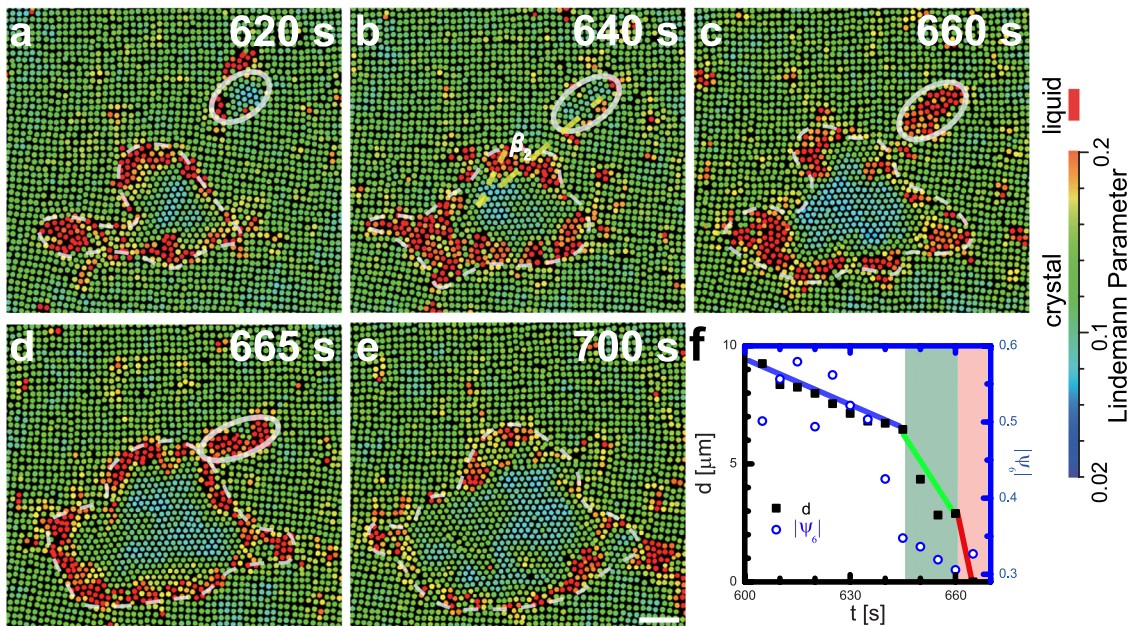

**Fig. 3 | Liquefication during nucleus coalescence. a, b** At $t = 620$ and 640 s, two nuclei grow independently. The misorientaiton angle between two nuclei $\beta_2 = 20°$ is shown by the two dashed lines in (**b**). The small nucleus circled by an ellipse transforms to a liquid nucleus at $t = 660$ s in (**c**); it is attracted to the large nucleus in (**d**) and merges with the large nucleus at $t = 700$ s in (**e**). **f** The evolution of separation $d$ between the two nuclei and six-fold orientational order parameter $|\psi_6|$ of the small nucleus during the liquefaction process. The colours in (**a–e**) represent the dynamic Lindemann parameter for each particle. Scale bar: 5 μm.

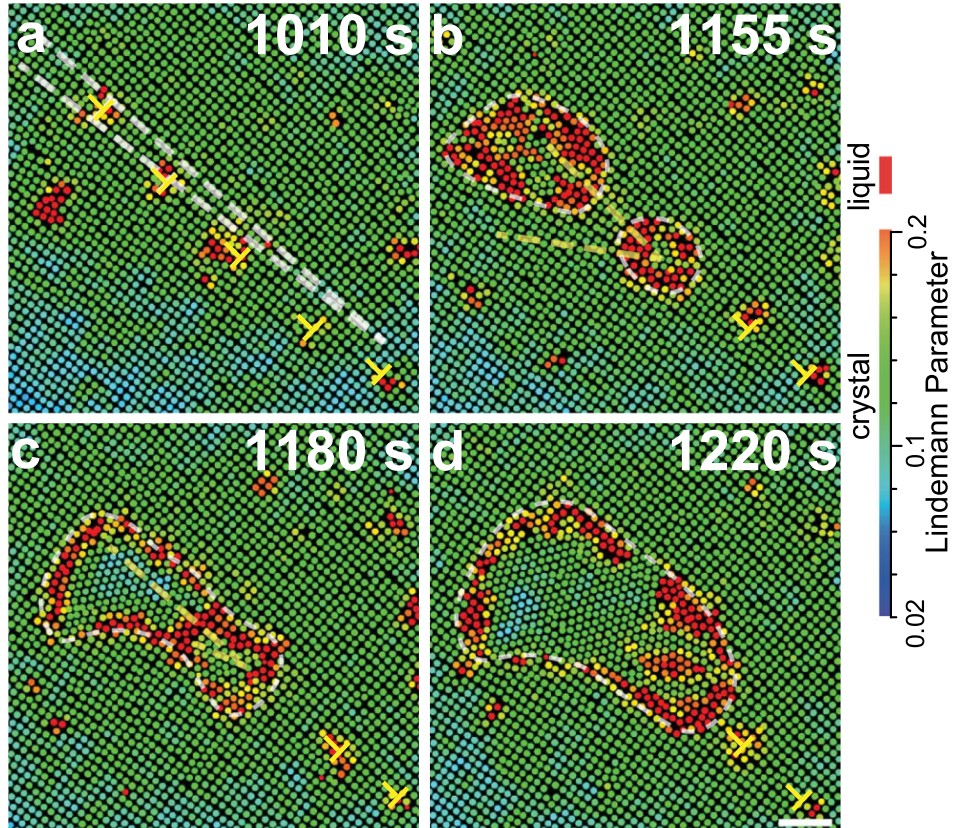

**Fig. 4 | Nucleus coalescence on a low-angle GB. a** The lattice directions of the two grains have a 5° misorientation angle labelled by the two white dashed lines. These two grains form a low-angle GB on their interface, which can be viewed as a chain of dislocations (⊥). **b** After an incubation time of $t = 1160$ s, two nuclei form on the dislocations of the GB. They are liquid at an early stage and then grow into triangle lattices in (**c**). Yellow dashed lines mark the lattice directions of two nuclei. **d** The two nuclei grow along the GB and merge. The colours in (**a**–**d**) represent the dynamic Lindemann parameter for each particle. Scale bar: 5 μm.

Supplementary Mov. 2, the two △ lattices with $\beta_2 = 20°$ do not coalesce and form a GB. Instead, the small nucleus melts when their separation is less than $10a$. The liquid can be identified from the amorphous structure shown in the raw images and active particle swapping in the video. Once the small nucleus becomes liquid, it migrates toward the large nucleus at a rate about one order magnitude faster than before (Fig. 3f). After the liquid nucleus merges into the large crystalline nucleus (Fig. 3d), it crystallizes and forms a single △ lattice without a GB (Fig. 3e). Such a liquefied small nucleus can be easily "swallowed" by the large crystalline nucleus owing to a lower free-energy barrier for the coalescence. The liquefaction can be understood as follows. The crystal-liquid interfacial energy is usually smaller than that of a c-c interface in colloidal crystals[12], metals, and alloys[1]. Thus, a smaller nucleus tends to melt because surface energy dominates over the bulk chemical potential. Moreover, liquid is able to relax stresses more efficiently near the large growing nucleus and during the coalescence.

In addition, small liquid nuclei can directly form on dislocations in the parent lattice near a large △-lattice nucleus. They are elongated and attracted towards the large △-lattice nucleus before they coalesce and recrystallise into the △ lattice (Supplementary Fig. 5).

### Coalescence of nuclei on a grain boundary

GBs and triple junctions can reduce the nucleation energy barrier. Thus, nuclei often form on these defects. We find that the coalescence of such nuclei is along the GB, as shown in Figs. 4, 5 for low- and high-angle GBs, respectively.

Type (4): Figure 4 shows that two nuclei emerge from two dislocations on a low-angle GB (Supplementary Mov. 3). Each exhibits

two-step diffusive nucleation with liquid intermediate state: □ → liquid → △. Since the recrystallized △ lattice in the liquid has a broad probability distribution of orientations, the △ lattices of the two nuclei have large $\beta_2$. They coalesce via a liquid interface which migrates to the small nucleus. During the coalescence, the small nucleus rotates to align with the large one; the large one barely reorients.

Type (5): Figure 5 shows nucleus coalescence on a high-angle GB (Supplementary Mov. 4). The rectangular nucleus on the GB has one coherent and three incoherent facets (Fig. 5a). At the nearby symmetric triple junction with three 120° angles, another nucleus is observed with triangle shape such that all its surfaces can be low-energy coherent facets. This nucleus grows very slowly because coherent facets are very stable[12]. For the rectangular nucleus on the GB, its left end near the asymmetric triple junction slowly grows wide, whereas the right end rapidly grows along the GB toward the small nucleus. As a result, an elongated triangular shape is produced (Fig. 5c). The nucleus grows faster along the GB because the GB is more disordered, which reduces the free-energy barrier of the diffusive transformation. The approach speed of the two nuclei increases when $d < 10a$ (Fig. 5f). When $d < 3a$, a small liquid channel develops between the two nuclei and then recrystallizes, thereby fusing the two △-lattice nuclei.

### Mechanisms of fast nucleus growth

All five types of nucleus coalescence scenarios in Figs. 1–5, Supplementary Figs. 3–5 exhibit three stages. (i) The two nuclei approach each other cooperatively when their separation is smaller than a critical distance $10 \pm 5a$. (ii) The nuclei are connected by a channel of the product phase, forming a dumbbell shape. (iii) The merged dumbbell-

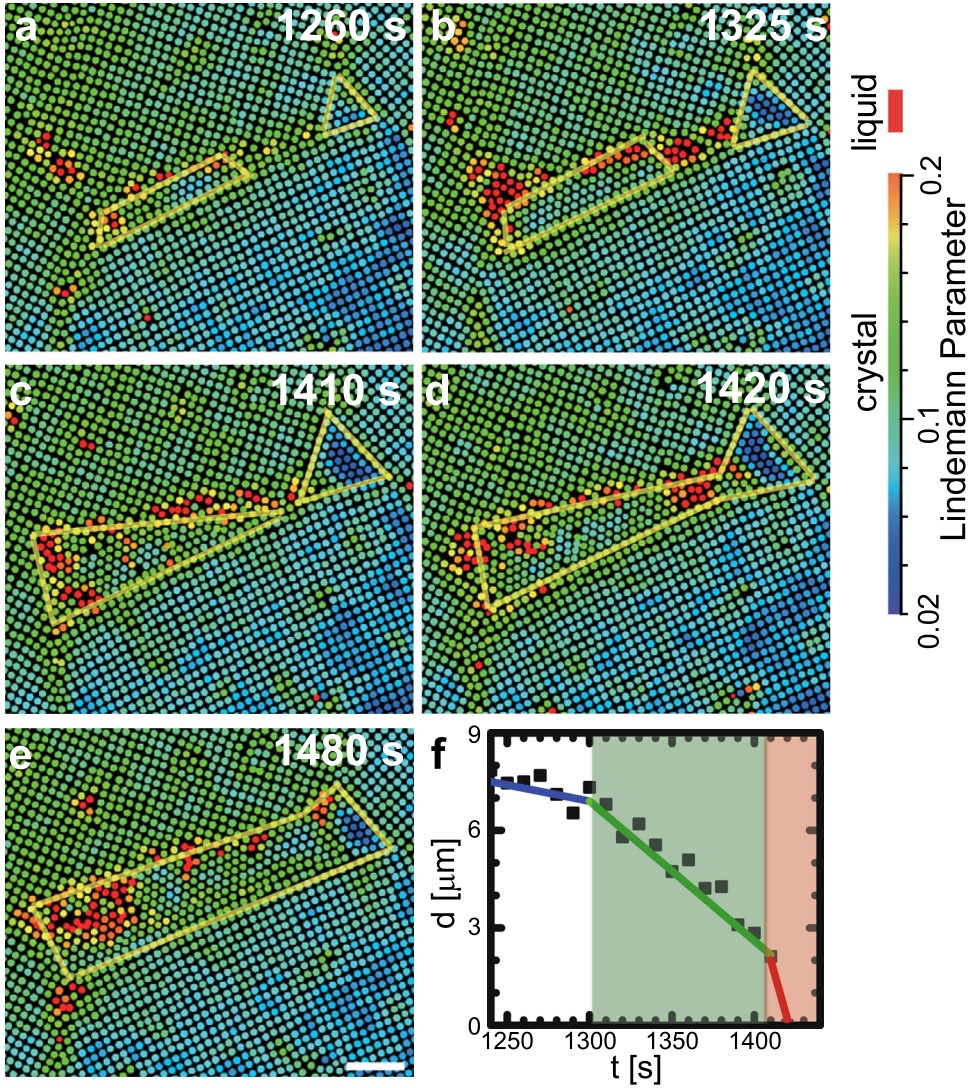

**Fig. 5 | Coalescence of nuclei on a high-angle (45°) GB. a** At $t = 1260$ s, notice a rectangular nucleus on a GB and a triangular nucleus at a triple junction. **b** At $t = 1325$ s, both nuclei grow. **c** The nucleus on the GB grows towards the triple junction and becomes triangular in shape. **d** Two nuclei merge together. **e** The nucleus becomes rectangular after coalescence. The colours in (**a**–**d**) represent the dynamic Lindemann parameter for each particle. Scale bar: 5 $\mu$m. **f,** The evolution of the separation between two nuclei.

shaped nucleus relaxes to a convex shape. Nuclei grow faster during all three stages; this faster growth is evident in the larger slopes of the three coloured regions, i.e., compared to the white regions (see Fig. 6a).

In stage (i), the two nuclei distort the parent lattice between them, leading to a faster growth of two nuclei thereafter. This phenomenon can be described by the general energy-barrier-crossing process sketched in Fig. 6b[39,40]. For this process, a particle's transformation rate from state A to state B is $fe^{-Q/k_BT}$, where $f$ is the collision frequency to jump over the free-energy barrier with height $Q$ separating states A and B (Fig. 6b). Similarly the transformation rate from B to A is $fe^{-(Q+\Delta\mu)/k_BT}$, where $\Delta\mu$ is the free-energy difference between states B and A. For a particle residing at the nucleus surface, A and B represent the 5□ and 4△ lattices, respectively. Hence, the net growth rate of a nucleus is proportional to

$$v \propto f(e^{-Q/k_BT} - e^{-(Q+|\Delta\mu|)/k_BT}). \qquad (1)$$

After the □ lattice becomes distorted, the system state is changed from A to an excited state A'. Since the chemical potential of a distorted lattice is higher, $|\Delta\mu'| > |\Delta\mu|$, and $Q' < Q$ as shown

in Fig. 6b. Consequently, the net growth rate becomes $v' \propto f(e^{-Q'/k_BT} - e^{-(Q'+|\Delta\mu'|)/k_BT}) > v$. We directly observed this faster growth. For example, the growth rate increases slightly when nuclei on GBs cause slight distortions, but the growth rate increases significantly when nuclei cause strong distortions inside a crystalline domain. Generally, we can expect these phenomena to occur because Eq. (1) is system independent and volume changes are ubiquitous in c-c transitions.

Interface energy is crucial in nucleus coalescence. It explains the faster growth in stages (ii) and (iii). In stage (ii), two nuclei rapidly link together, when their separation is less than approximately $3a$, by forming a channel of liquid or product lattice. Formation of such channels is free-energy favourable, because the volume growth of the product phase does not strongly increase surface area. In stage (iii), the concave nucleus grows faster near its negative curvature; it becomes convex to reduce the interfacial energy. By contrast, nanocrystalline platinum nuclei in a supersaturated liquid solution coalesce and then partly dissolve to a smaller size[23]. We attribute this difference to the following. In c-c transitions, there are sufficient particles in the parent phase to form the product phase, i.e., the process is reaction-controlled. However, crystallization in a liquid solution may result in

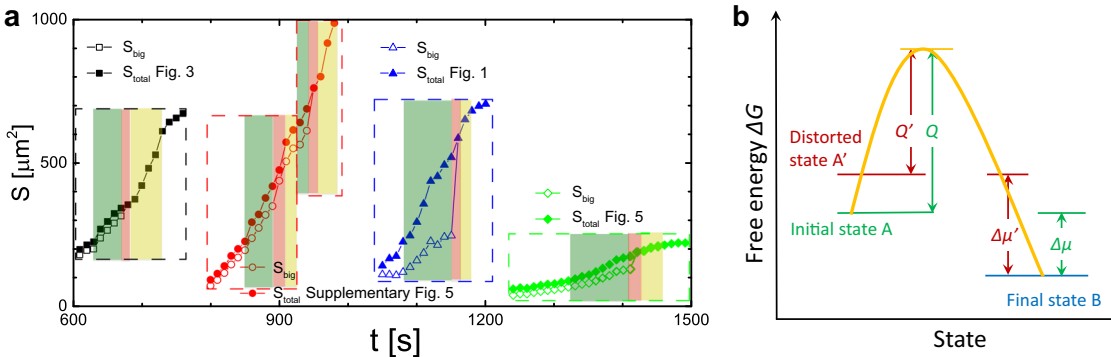

**Fig. 6 | Crystal growth during nucleus coalescence. a** The evolution of the nucleus area $S$ for the larger nucleus (open symbols, $S_{big}$) and for the sum of two nuclei (solid symbols, $S_{tot}$) for Fig. 3 (black square), Supplementary Fig. 5 (red circle for two coalescence events), Fig. 1 (blue triangle), and Fig. 5 (green diamond). The green, red, and yellow shaded regions represent stages (i), (ii) and (iii), respectively; the slopes are larger in these regions compared to the non-shaded regions. **b** Schematic of barrier-crossing processes for a normal (state A) and distorted (state A') square lattices to the product triangle lattice.

**Table 1 | Nucleus coalescence kinetics in the 5square → 4triangle transition**

| Type | 1 | 2 | 3 | 4, 5 |
|---|---|---|---|---|
| Location | Inside domain | Inside domain | Inside domain | On low- or high-angle GB |
| Nucleation mechanism | Marternsitic | Diffusive | Martensitic and diffusive | Diffusive |
| Lattice orientation | Parallel | Angled | Angled | Random |
| Nucleus coupling | Via distortion of parent lattice | Via distortion of parent lattice | Small one melts accompanied by movement | By growth through GB |
| Example | Fig. 1 | Fig. 2 | Fig. 3 | Fig. 4 or 5 |

The nucleation mechanisms were previously reported in ref. 12,13. In addition to the five types of coalescence between two crystalline nuclei, coalescence also occurs between a crystalline nucleus and a liquid one(Supplementary Fig. 5), which has three stages similar to those shown in Fig. 6.

a depleted region near the nucleus so that particles need time to diffuse over distance to attach onto the nucleus, i.e., the process is diffusion-controlled[41].

The kinetic pathways of coalescence processes are summarized in Table 1. Type (1) usually occurs in martensitic nucleation which possesses a special angle between the product and parent lattices[13]. Martensitic nucleation occurs within crystalline grains under external stresses. By contrast, diffusive nucleation occurs both inside grains and on GBs under no external stresses and often at low volume fractions. It generates a broad distribution of lattice orientations[12], yielding unparalleled lattice orientations of the two nuclei. Type (2), (4) and (5) occur in diffusive nucleation. Softer parent lattices featured with larger Lindemann parameters promote diffusive nucleation (i.e. random misorientation angle between two nuclei) and the liquefaction. Defects, especially GBs, usually reduce the free energy barrier of nucleation. Type (5) is expected to occur more frequently than type (2) and (4). The small nucleus in type (3) forms via the martensitic transformation in the early stage, and then move diffusively after liquefication. This occurs in a narrow regime around the boundary of martensitic and diffusive regimes in the parameter space of volume fraction and stress[13], thus type (3) has a low probability.

## Discussion

Using high-quality thin-film colloidal crystals with diameter-tunable microspheres and local optical heating, we are able to image the microscopic dynamics of nucleus coalescence in c-c transitions. All five types of nucleus-coalescence scenarios (Figs. 1–5, Supplementary Figs. 3–5) exhibit three stages: they approach each other, they attach, and the shape of the coalesced nucleus relaxes. These qualitative coalescence behaviours are reproduced in tens of experimental trials. In total the experimental work reveals six phenomena that are difficult to resolve in atomic crystals. First, all three stages of the coalescence enhance the nucleus growth rate. Second, the volume change caused by nucleus growth distorts the parent lattice between the two nuclei

when the nucleus separation is less than ~10$a$. Such strain effectively attracts the growth fronts of the two nuclei and promotes their growth. Third, the coalesced nucleus grows faster in the concave regions. Fourth, a small △-lattice nucleus with a large misorientation angle liquefies as it approaches a large nucleus, which promotes its mobility and coalescence speed. The liquefaction results from the strain in parent lattices and lower liquid-solid interfacial energy. Fifth, two crystalline nuclei with different lattice orientations coalesce and form a GB in the product phase. The high-angle GB sweeps through the small nucleus whereas the low-angle GB dissociates into dislocations which diffuse apart. Finally, small nuclei move and rotate more easily than large ones during the coalescence.

Here we have observed these phenomena in crystals composed of micrometer sized colloidal particles. Nevertheless, we expect that many of these effects similarly exist in atomic crystals, because volume-change-induced strain, minimisation of interfacial energy, and barrier-crossing processes are universal in c-c transitions. For example, a density difference between the parent and product lattices exists in both colloidal and atomic crystals and induces similar strain fields affecting the nucleus coalescence. Thus, the faster nucleus growth in stage (i) should generally occur. In a different vein, crystal-liquid interfaces usually have lower surface tension than c-c interfaces in colloidal crystals[12], metals, and alloys[1]; thus, liquefaction of a small crystalline nucleus during the coalescence can be expected to generally occur. Indeed, a single metastable liquid nucleus has been reported in experiments on graphite-diamond transitions[42] and crystal-amorphous transitions in three dimensional (3D) metals[21], and has been observed in simulations of c-c transitions of 2D ice[43] and hard-sphere crystals[5]. However, nucleus coalescence induced liquid has not been reported. After two nuclei are connected, the concave region of the coalesced nucleus grows faster, thereby relaxing the nucleus shape to become convex for lower interfacial energy. In addition, systems with complex pair interactions, such as colloidal crystals composed of nanometer-sized atomic/molecular crystallites, can form more types of lattices[44]. They may exhibit richer

nucleation and coalescence behaviours in crystal-crystal transitions. Our results in thin films can be generalised to 3D systems because the nucleation in both types of systems are governed by competition between the surface energy term and bulk chemical potential term[12]. This connection is important because nucleus coalescence affects phase-transition rates, defect generation, and the mechanical properties of the product crystal. Thus, our elucidation of microscopic kinetics could help with control of the microstructure and material properties of crystalline matter at atomic, micro- and mesoscale.

## Methods
### Sample preparation
We synthesise NIPA microgel spheres and disperse them in an aqueous buffer solution with 1 mM acetic acid. They are slightly charged with short-range steric repulsions[28,45]. We measure the effective diameter $\sigma$ by direct imaging of isolated particles stuck to the glass wall in a dilute suspension; $\sigma$ decreases linearly with increased temperature $T$ (Supplementary Fig. 1a)[29]. To avoid ambiguity in the definition of diameter for soft spheres, the measured $\sigma(T)$ is rescaled such that the melting volume fraction of the 3D crystal is equal to that of hard spheres ($\phi_m^{3D} = 54.5\%$). The measured freezing point ($\phi_f^{3D} = 49\%$) is very close to that of hard spheres 49.4%, which indicates that the phase behaviours of NIPA colloids are reasonably well modeled by hard spheres. According to the phase diagram obtained by simulations[30,31], hard spheres confined between two hard walls exhibit $n\square \to (n-1)\triangle$ but not $n\square \to n\triangle$ or $n\triangle \to n\square$ transition by decreasing $\sigma$, i.e., increasing the effective temperature, as confirmed in the experiment.

A droplet of colloidal suspension is directly added between two glass walls. Colloidal particles form polycrystals with typical domain size ranging within $10-300\,\mu m$. The sample cell thickness $H$ is roughly controlled by the added volume of colloidal suspension before we seal the sample and fix its thickness. The refractive indices of water and NIPA spheres are very close, because over 90% of the microgel is water. Consequently, bulk crystalline layers can be imaged clearly even with bright-field microscopy. The images become blurry when spheres form a disordered liquid. In most experiments, we monitor a surface layer because particles in liquid regions can be imaged clearly, and particles in $n\square$ and $(n-1)\triangle$ are in the same focal plane. Before the experiment, we use the temperature controller to cycle the temperature slowly below the transition point to anneal out small defects and release possible pressure that may have built up during colloid filling.

### Local heating
We locally heat the interior of the crystals across the c-c transition point by using a beam of light from a 100 W mercury lamp to illuminate a portion of the sample, while the ambient lattice temperature is held below the transition point (Supplementary Fig. 2). Otherwise, nuclei usually form outside the field of view and become very large when they grow into the field of view. Local heating also enables us to produce homogeneous or heterogeneous nucleation by choosing the heated area in a defect-free region or near a GB, respectively. The local heating area in the focal plane can range from $20\,\mu m$ to $5\,mm$ in diameter by tuning the iris; it is usually set to $76\,\mu m$. The local heating area is observed in the transmission mode of the optical microscope to avoid direct exposure of the camera to heating light. A dilute non-fluorescent black dye (Chromatech-Chromatint black 2232 liquid), 0.6% by volume, is added to absorb the heating light. To the best of our knowledge, the dye did not change the phase behaviour. A paraffin film is placed in the light path to make the temperature profile uniform.

Supplementary Fig. 1b shows that the 2.0 °C heating effect can quickly stabilise 2 s after the light is turned on. The heating effect is measured from $\delta T = T_m - T_m^h$, where $T_m^h$ and $T_m$ are the melting temperatures at a GB with and without the optical heating, respectively. $\delta T$ depends on the light intensity and the dye concentration, and is usually set to 2.0 °C. The heating profile shown in Supplementary

Fig. 2b is measured from a $5\,\mu m$ thick aqueous solution of yellow fluorescein (0.01% by weight). The brightness of the fluorescent solution, the light intensity, and the heating effect are linearly dependent in ref. 46. The light from mercury lamp is focused by a 100 × objective, so the heating effect is strongest at the focal plane. By measuring the melting point of 3D NIPA colloidal crystals, we find that the temperature variation is less than 0.2 °C in ± 25 layers along the z-axis, so the temperature is uniform enough across the z direction in a five-layer thick sample. Indeed, no difference is observed between the transition behaviours at the top and the bottom walls.

### Data analysis
The Lindemann parameter is defined as the ratio of the vibration magnitude of a particle around its equilibrium position in a crystalline lattice to the lattice constant. In quasi-2D or thin-film systems, the Lindemann parameter diverges slowly even in crystals. Thus, we define the dynamic Lindemann parameter as $L = \sqrt{\frac{\langle [\Delta \mathbf{x}_j(t) - \Delta \mathbf{x}_k(t)]^2 \rangle}{2a^2}}$ [47], where $\Delta \mathbf{x}_j$ is the displacement of particle $j$. Particles $k$ and $j$ are nearest neighbours determined by Delaunay triangulation. $L$ usually reaches a plateau within $t = 1\,s$, which is used to label the colours of particles.

We define the orientational order parameter of a particle as $\psi_{mj} = \Sigma_{k=1}^{n_j} e^{mi\theta_{jk}}/n_j$, where $i = \sqrt{-1}$. $m = 4$ and 6 corresponds to four- and six-fold orientational order parameters, respectively. The nearest neighbours in $\triangle$ lattices are directly obtained from Delaunay triangulation, which yields $\langle n_j \rangle = 6$. However, a particle in $\square$ lattices usually has four nearest neighbours and Delaunay triangulation incorrectly takes about two second nearest neighbours as nearest neighbours. Thus, we further limit the nearest neighbours to the particles whose distance is less than $1.2a$. An $m$-fold crystalline bond linking particle $j$ and $k$ is defined as $|\psi_{mj}^* \psi_{mk}| \geq 0.5$. A particle in a $\square$ or $\triangle$ lattice is defined as one with > 2 four-fold crystalline bonds or > 3 six-fold crystalline bonds, respectively. Otherwise, the particle is in the liquid phase. Our main results are not sensitive to the threshold changes and the liquid-like particles are confirmed by Lindemann parameter and particle swapping in the recording videos. A nucleus is defined as a cluster consisted of particles with low four-fold order parameters and are connected by nearest-neighbour bonds.

## Data availability
All data that support the plots within this manuscript are available in Figshare [https://doi.org/10.6084/m9.figshare.22774814].

## Code availability
The codes that analyse dynamic Lindemann parameters and orientational order parameters are available upon request.

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

## Acknowledgements

This work was supported by the National Natural Science Foundation of China 12074406 and T2221001(Y.P.), Research Grants Council C6016-20G(Y.H.), and Natural Science Foundation DMR2003659 and MRSEC DMR1720530(A.G.Y.). We thank Xian Chen for helpful discussions.

## Author contributions

Y.P. and Y.H. designed the research; Y.P. performed the experiment; T.S. and A.G.Y. provided the colloid; Y.P. analyzed the data with the help from W.L. and Y.H.; and Y.P., A.G.Y., and Y.H. wrote the paper.

## Competing interests

The authors declare no competing interests.
