## [Peer Review File · Nature Communications]

In situ observation of coalescence of nuclei in colloidal crystal-crystal transitionsREVIEWER COMMENTS

Reviewer #1 (Remarks to the Author):

In the manuscript by Peng et al., the authors investigate the coalescence of two triangular lattices nucleated from a square-triangular solid-solid transition. They observe five different types of kinetic pathways in situ using optical microscopy with a single-particle resolution. The pathway for two nuclei to coalesce depends on their relative lattice orientation, size, as well as the location where they nucleate (inside the bulk or near the grain boundary). Despite the variety of coalescence pathways, one common feature they share is that the growth of nuclei accelerates. To rationalize this observation, they provide an assumption based on how distortion caused by nuclei coalescence may reshape the free energy landscape between the nucleated phase and the parent phase.

The images and videos the authors provide clearly reveal the real-space trajectory of nuclei coalescence. The study is carried out meticulously with a detailed classification of pathways. Their findings are novel and interesting and may shed light on the fundamental comprehension of the solid-solid transitions in atomic crystals. I recommend a publication of this manuscript after minor revisions. My questions are listed below:

1) In Fig.1, as the surfaces of two triangular lattices approach each other, some defects seem to appear from the parent phase between the two grains. As the connected region grows, these defects seem to be either pushed to the surface and expelled, or absorbed by the bulk. Can the authors elaborate on the dynamics of defects caused by nuclei coalescence and maybe provide some insights about how these defects are formed and annealed?

2) The parent grain in Fig.2 seems to be less ordered than the single-crystal grain in Fig.1. How does the quality of the parent crystal affect the coalescence of nuclei, as well as the grain boundary formed during the course of coalescence?

3) I noticed that the Lindemann parameter in Fig.3&4 is higher than that in Fig.1&2. This means the modulus of the parent phase is lower in Fig.3&4. Could the difference in the softness (or the degree of superheating) of the parent phase be part of the reason for different kinetic pathways?

4) Considering the variety of the discovered pathways, one suggestion for the authors is that they could use a table to relate the features of different pathways to different scenarios, which helps readers to easily capture the key findings.

And a typo:

1) In Line 105, 'Fig.S3c' is nowhere to find. It should be Fig.S1c presumably.

Reviewer #2 (Remarks to the Author):

In this article, the authors described the coalescence of nuclei using colloidal particles as model systems to investigate crystal-crystal transitions, in particular the coalescence during the crystal-crystal (c-c) phase transition. Based on their previous endeavors on colloidal phase transition, high-quality experimental setup of thin-film colloidal particle deposition, and

ability to achieve local heating through beam of light, the authors demonstrated five types of kinetic pathways of nuclei coalescence, indicating the complex process that varies case-by-case due to different nucleus sizes, interfaces, and lattice orientation. Finally, the authors explained the underlying mechanisms behind the phenomena, which can be ascribed to the reduction of interface energy. Overall, this is a thorough study showing the detailed dynamics of nuclei coalescence during c-c phase transformation for the first time. The discussion follows some previous conclusions of this system (same first author, Nature Communications 8, 14978, 2017; Nature Materials 14, 101, 2015), and the reasoning behind those new phenomena (strain from volume change, interfacial energy minimization, and barrier-crossing processes) can also be generalized into atomic crystals. I would recommend its publication in Nature Communications once they address the following minor issues.

1. According to the experimental setup, the authors used a beam of light to heat local particles within the field of view to induce phase transition and observed five types of coalescence pathways. Which of the five pathways are dominant? Are there ways to control or realize one pathway over the others? It would be better if the authors can present the statistics of the pathways and at least provide some discussions about the pathway control or selection.
2. There is only one case of type 1 (Fig. 1) pathway based on the alpha angle analysis (distortion to parent lattice in between two coalescing nuclei). Are there more examples to support this distortion idea?
3. There is only one case (Fig. 3) that shows a nucleus melting case. Is there more data to support such a liquefaction pathway?
4. The definition of beta has changed. Line 132 denotes beta as the misorientation angle between nucleus and parent lattice; Line 176 (and all beta afterward) denotes beta as the misorientation angle of two nuclei. Better to set 2 symbols for them.
5. It is difficult to identify triangle and square lattices when they are only colored by Lindemann parameter. The authors can try to label the boundaries to make the differences clear, especially in Fig. 2.
6. Missed the description of analysis details: Lindemann parameter, the definition of particles that belong to a nucleus;
7. Line 178 discusses "dislocations have effective repulsion". Not clear about what this means without some elaborations or references.
8. The author has already compared on the similarities and differences between colloidal systems and atomic crystals. It would be great if the authors can include some discussion of the in-situ study on nanoparticle superlattice and even give some outlook in the conclusion part.

Reviewer #3 (Remarks to the Author):

The authors use light microscopy to study the temperature-induced transformation of colloidal crystals between square and triangular lattice forms. They observe various mechanisms of transformation including the appearance and merging of multiple nuclei. They study the rates of growth and coalescence of nuclei, identifying regimes of independent and cooperative growth, and situations in which nuclei exhibit liquidlike structure prior to solidification.

I found the paper clear and interesting. The question of how materials change phase is an

important one, and in recent years there have been a number of observations of interesting “nonclassical” mechanisms that proceed in a way that is different to the appearance and growth of single, ordered nuclei. The present paper clearly illustrates a nonclassical pathway in which nuclei coalesce. While similar observations have been made before in e-beam microscopy, which has revealed nanoparticle assembly pathways involving coalescing clusters, the authors’ system seems to me to be a particularly clean one for the study of such mechanisms. I think the findings that have been presented are clearly described and interesting in their own right. I also think that the set up is more controlled than e-beam experiments (which tend to induce assembly in a way that is hard to control), and will encourage similar studies whose aim is to visualize phase-change pathways.

For these reasons I think the paper would interest the broad readership of Nat Comms. Before publication I would request that the authors consider the following points.

Figures. The snapshots of phase change are beautiful (e.g. Fig 1 a-f), but sometimes hard to follow in detail because of the color scheme and image size. I have great difficulty with subtle greens and blues and reds (e.g. Fig 4 — red on green is hard to see) at the magnification shown. Some possible alternatives: 1) The authors could consider a separate color scheme in the supplement. 2) Drawing lines between the centers of particles within some cutoff threshold would make triangles and squares easy to see; these shapes could be colored according to the average of the Lindemann parameter of the constituent particles. 3) Show images at the magnification of the SI movies, which are clearer.

Abstract — The connection of the final paragraph to the previous one is not entirely clear. Do those related mechanisms follow immediately from the stated observations? Given the inter-colloidal interactions, why would we expect a difference in growth mechanisms?

Line 98: Could state explicitly “for n within the range...”

Caption Fig. 3 — best to define beta in the caption. The color scheme here is particularly difficult. I’d suggest introducing the color bar so that the reader doesn’t have to go back to Fig. 1 (same comment for subsequent figs).

Panel 3f: given that the figure talks about liquefaction, can you show an order parameter that makes this clear?

Fig. 4 caption: best to defined GB.

Fig. 5: How are the lines drawn? By eye or according to an order-parameter threshold?

Fig. 6. The caption in panel a is hard to associate with individual lines — I suggest the captions be placed next to the relevant sub-panel. Also, how are the different regimes (shades) determined? Note typo in final line of that figure.

Can the parameters of Fig. (1) be estimated from data? Does the numerical enhancement of growth rate follow from those numbers?

REVIEWER COMMENTS

Response to Reviewer #1:

In the manuscript by Peng et al., the authors investigate the coalescence of two triangular lattices nucleated from a square-triangular solid-solid transition. They observe five different types of kinetic pathways in situ using optical microscopy with a single-particle resolution. The pathway for two nuclei to coalesce depends on their relative lattice orientation, size, as well as the location where they nucleate (inside the bulk or near the grain boundary). Despite the variety of coalescence pathways, one common feature they share is that the growth of nuclei accelerates. To rationalize this observation, they provide an assumption based on how distortion caused by nuclei coalescence may reshape the free energy landscape between the nucleated phase and the parent phase.

The images and videos the authors provide clearly reveal the real-space trajectory of nuclei coalescence. The study is carried out meticulously with a detailed classification of pathways. Their findings are novel and interesting and may shed light on the fundamental comprehension of the solid-solid transitions in atomic crystals. I recommend a publication of this manuscript after minor revisions. My questions are listed below:

1) In Fig.1, as the surfaces of two triangular lattices approach each other, some defects seem to appear from the parent phase between the two grains. As the connected region grows, these defects seem to be either pushed to the surface and expelled, or absorbed by the bulk. Can the authors elaborate on the dynamics of defects caused by nuclei coalescence and maybe provide some insights about how these defects are formed and annealed?

Reply: We thank the referee for the recommendation of publication.

As the surfaces of two triangular lattices approach each other, the parent square lattice between them slightly rotates as shown in Fig. 1h. In Line 153, we add “Such rotation may generate small defects such as two interstitials shown in Fig. 1c which quickly transform to the triangular lattices.” Since these defects are not very obviously and their lifetime is short, we did not mention them in our first submission. In the resubmission, we add two hollow stars in Fig. 1c to identify the interstitials and descriptions in the caption.

2) The parent grain in Fig.2 seems to less ordered than the single-crystal grain in Fig.1. How does the quality of the parent crystal affect the coalescence of nuclei, as well as the grain boundary formed during the course of coalescence?

Reply: The low quality of the parent crystal, i.e., more preexisted defects with their associated strains, can lower the nucleation barrier and facilitate the coalescence. The formation of a grain boundary during the coalescence mainly depends on the misorientation of two triangular lattices. The few preexisting defects appear not affecting much on the grain boundary formed during the coalescence.

3) I noticed that the Lindemann parameter In Fig.3&4 is higher than that in Fig.1&2. This means

the modulus of the parent phase is lower in Fig.3&4. Could the difference in the softness (or the degree of superheating) of the parent phase be part of the reason for different kinetic pathways?

Reply: Yes, we add the following sentence in Line 288 “Softer parent lattices featured with larger Lindemann parameters promote the diffusive nucleation (i.e. random misorientation angle between two nuclei) and liquefaction.”

4) Considering the variety of the discovered pathways, one suggestion for the authors is that they could use a table to relate the features of different pathways to different scenarios, which helps readers to easily capture the key findings.

Reply: Thanks, we add Table 1 and a paragraph to relate the features of different pathways to different scenarios in Pages 11-12.

And a typo:

1) In Line 105, ‘Fig.S3c’ is nowhere to find. It should be Fig.S1c presumably.

Reply: Yes, thank you. We correct this typo.

Response to Reviewer #2:

In this article, the authors described the coalescence of nuclei using colloidal particles as model systems to investigate crystal-crystal transitions, in particular the coalescence during the crystal-crystal (c-c) phase transition. Based on their previous endeavors on colloidal phase transition, high-quality experimental setup of thin-film colloidal particle deposition, and ability to achieve local heating through beam of light, the authors demonstrated five types of kinetic pathways of nuclei coalescence, indicating the complex process that varies case-by-case due to different nucleus sizes, interfaces, and lattice orientation. Finally, the authors explained the underlying mechanisms behind the phenomena, which can be ascribed to the reduction of interface energy. Overall, this is a thorough study showing the detailed dynamics of nuclei coalescence during c-c phase transformation for the first time. The discussion follows some previous conclusions of this system (same first author, Nature Communications 8, 14978, 2017; Nature Materials 14, 101, 2015), and the reasoning behind those new phenomena (strain from volume change, interfacial energy minimization, and barrier-crossing processes) can also be generalized into atomic crystals. I would recommend its publication in Nature Communications once they address the following minor issues.

Reply: We thank the referee for the recommendation of the publication.

1. According to the experimental setup, the authors used a beam of light to heat local particles within the field of view to induce phase transition and observed five types of coalescence pathways. Which of the five pathways are dominant? Are there ways to control or realize one pathway over the others? It would be better if the authors can present the statistics of the pathways and at least

provide some discussions about the pathway control or selection.

Reply: We appreciate this insightful question. The probabilities of pathways depend on the sample [volume fraction, defect density, nucleation mechanism (martensitic or diffusive which affects the angle between the parent and product lattices)] and the selected heated region (within crystalline domain or on grain boundaries). We add Table 1 and its related paragraph in Pages 11-12 to summarize the pathway selection of coalescence.

2. There is only one case of type 1 (Fig. 1) pathway based on the alpha angle analysis (distortion to parent lattice in between two coalescing nuclei). Are there more examples to support this distortion idea?

Reply: Yes, type 1 has been observed in many of our samples and all of them exhibit the distortion. The distortion is revealed by the analysis of the angle α and orientational order parameter $|\psi_4|$. As another example of type 1, Fig. S4 also shows a distortion, although it is smaller ($\approx 7^\circ$) than that in Fig. 1 due to the coherency of the triangular-square interfaces.

3. There is only one case (Fig. 3) that shows a nucleus melting case. Is there more data to support such a liquefaction pathway?

Reply: The liquefaction event is rare compared with other types of coalescence because it occurs when both diffusive nucleation and martensitic nucleation can occur in one sample. Fig.5a of Ref. 13 shows that it is a narrow regime in the parameter space of the external stress (or flow) and the volume fraction (or Lindemann parameter). Moreover, the two nuclei should have a large misorientation angle. These requirements make this pathway rare.

4. The definition of beta has changed. Line 132 denotes beta as the misorientation angle between nucleus and parent lattice; Line 176 (and all beta afterward) denotes beta as the misorientation angle of two nuclei. Better to set 2 symbols for them.

Reply: We thank the referee for the suggestion. We distinguish these two angles by naming the misorientation angle between a nucleus and the parent lattice as β_1 and the misorientation angle of two nuclei as β_2 .

5. It is difficult to identify triangle and square lattices when they are only colored by Lindemann parameter. The authors can try to label the boundaries to make the differences clear, especially in Fig. 2.

Reply: The field of view in Fig. 2 needs to be large to include all nuclei, which makes the lattice orientations less clear. We label the boundaries in Fig. 2 as the reviewer suggested.

6. Missed the description of analysis details: Lindemann parameter, the definition of particles that belong to a nucleus;

Reply: Thanks, we add the descriptions of analysis details in Methods (Lines 400-422).

7. Line 178 discusses "dislocations have effective repulsion". Not clear about what this means without some elaborations or references.

Reply: An edge dislocation can be viewed as inserting an extra half plane of particles which is associated with a long-range strain field and additional elastic energy. The strain energy is characterized by a Burgers vector and the elastic energy is proportional to the square of the Burgers vector. A dislocation interacts with other dislocations and grain boundaries due to their overlapped strain field. Thus, the movements and interactions of dislocations are complicated and are not the focus of this study. We add Ref. 38 (*Introduction to dislocations* by D. Hull and D. J. Bacon) and change the text to "... which can be viewed as a chain of dislocations whose Burgers vectors are parallel (Fig. 2b). These dislocations move under thermal fluctuations and elastic stresses in the lattice [38]. One dislocation falls into the nearby GB to minimize the elastic energy (Fig. 2c, d), ..."in Line 180.

8. The author has already compared on the similarities and differences between colloidal systems and atomic crystals. It would be great if the authors can include some discussion of the in-situ study on nanoparticle superlattice and even give some outlook in the conclusion part.

Reply: Colloidal crystals composed of nm-sized atomic/molecular crystallites sometimes are called as nanocrystal superlattices (e.g. BNSLs for binary nanocrystal superlattices); However, atoms/molecules in each building block (colloidal particle) forming a lattice or amorphous solid is irrelevant for the colloidal crystal. We assume that the "nanoparticle superlattice" here refers to such system. We add "In addition, systems with complex pair interactions, such as colloidal crystals composed of nanometer-sized atomic/molecular crystallites, can form more types of lattices [44]. They may exhibit richer nucleation and coalescence behaviours in crystal-crystal transitions." in Line 315.

Response to Reviewer #3:

The authors use light microscopy to study the temperature-induced transformation of colloidal crystals between square and triangular lattice forms. They observe various mechanisms of transformation including the appearance and merging of multiple nuclei. They study the rates of growth and coalescence of nuclei, identifying regimes of independent and cooperative growth, and situations in which nuclei exhibit liquidlike structure prior to solidification.

I found the paper clear and interesting. The question of how materials change phase is an important one, and in recent years there have been a number of observations of interesting "nonclassical" mechanisms that proceed in a way that is different to the appearance and growth of single, ordered nuclei. The present paper clearly illustrates a nonclassical pathway in which nuclei coalesce. While similar observations have been made before in e-beam microscopy, which has revealed nanoparticle assembly pathways involving coalescing clusters, the authors' system seems to me to be a particularly clean one for the study of such mechanisms. I think the findings

that have been presented are clearly described and interesting in their own right. I also think that the set up is more controlled than e-beam experiments (which tend to induce assembly in a way that is hard to control), and will encourage similar studies whose aim is to visualize phase-change pathways.

For these reasons I think the paper would interest the broad readership of Nat Comms. Before publication I would request that the authors consider the following points.

Figures. The snapshots of phase change are beautiful (e.g. Fig 1 a-f), but sometimes hard to follow in detail because of the color scheme and image size. I have great difficulty with subtle greens and blues and reds (e.g. Fig 4 — red on green is hard to see) at the magnification shown. Some possible alternatives: 1) The authors could consider a separate color scheme in the supplement. 2) Drawing lines between the centers of particles within some cutoff threshold would make triangles and squares easy to see; these shapes could be colored according to the average of the Lindemann parameter of the constituent particles. 3) Show images at the magnification of the SI movies, which are clearer.

Reply: Thank you for the suggestions. Both triangular and square lattices contain strongly and weakly vibrating particles, thus their colour ranges are similar. Triangular and square lattices can be directly seen from the structure in the raw image even without colour. The magnified image cannot include all the related nuclei, e.g. in Fig. 2. To better distinguish the two lattices, we follow the suggestion of Reviewer 2 to add dashed curves on their interfaces.

Abstract — The connection of the final paragraph to the previous one is not entirely clear. Do those related mechanisms follow immediately from the stated observations? Given the inter-colloidal interactions, why would we expect a difference in growth mechanisms?

Reply: Yes, the mechanisms listed in the final paragraph are based on the general behaviours of crystals (not depending on particle interactions), thus the observations in the previous paragraph can be expected in atomic system as explained in the final paragraph. We do not expect very different growth mechanisms for various crystals although detailed kinetics can be different due to factors such as effective temperature (volume fraction), defect density, lattice symmetry, interface coherency, etc..

Line 98: Could state explicitly “for n within the range...”

Reply: We add ‘(n = 5, 6)’ for our samples in Line 98.

Caption Fig. 3 — best to define beta in the caption. The color scheme here is particularly difficult. I’d suggest introducing the color bar so that the reader doesn’t have to go back to Fig. 1 (same comment for subsequent figs).

Reply: Besides the definition of β in the text, we repeat the definition in the caption for the easier understanding. We rename β as β_2 to distinguish it from β_1 .

Panel 3f: given that the figure talks about liquefaction, can you show an order parameter that makes this clear?

Reply: The liquid can be directly seen from the amorphous structure in the image and the active particle swapping motions in the video. We add “The liquid can be identified from the amorphous structure shown in the raw images and active particle swapping in the video.” in Line 199. We further add data and descriptions for the six-fold orientational order parameter $|\psi_6|$ of the region around the small nucleus in Fig. 3f and Line 149, respectively.

Fig. 4 caption: best to defined GB.

Reply: We revise the second sentence of Fig. 4(a) caption to define a low-angle GB: “These two grains form a low-angle GB on their interface, which can be viewed as a chain of dislocations (\perp)”.

Fig. 5: How are the lines drawn? By eye or according to an order-parameter threshold?

Reply: We draw the lines by eyes because the interfaces are obvious to tell. We tried to draw them according to the orientational order parameter, which is similar to the current one but very noisy.

Fig. 6. The caption in panel a is hard to associate with individual lines — I suggest the captions be placed next to the relevant sub-pabel. Also, how are the different regimes (shades) determined? Note typo in final line of that figure.

Reply: We move the legends to their corresponding subpanels in Fig. 6a.

The green, red and yellow shaded regimes represent stages (i), (ii) and (iii), respectively. How these stages are determined is reported in Lines 242-245. In stage (i), the nucleus growth is faster than its initial isolated state (green lines in Figs. 1g, h, 3f, 5f). In stage (ii), the nearest interfaces of two nuclei connect via a narrow channel (red lines in Figs. 1g, h, 3f, 5f). Stage (iii) is defined as the relaxation of nucleus shape from concave to convex. We change the caption “The green, red, and yellow shaded regions represent stages (i), (ii) and (iii), respectively”.

We thank the referee and correct the typo in the final line.

Can the parameters of Fig. (1) be estimated from data? Does the numerical enhancement of growth rate follow from those numbers?

Reply: Yes, we calculated the parameters of Fig.1, including r , d , $|\psi_4|$, α and growth rate from the image analysis of the raw video. Yes, the enhancement of growth rate follows from those numbers.

REVIEWERS' COMMENTS

Reviewer #1 (Remarks to the Author):

I am satisfied with the answers of all questions and suggest an acceptance. A minor suggestion: The words in Tab.1 need to be aligned well.

Reviewer #3 (Remarks to the Author):

The authors have addressed my comments from the previous round of review, and I recommend publication.

Reviewer #1 (Remarks to the Author):

I am satisfied with the answers of all questions and suggest an acceptance. A minor suggestion: The words in Tab.1 need to be aligned well.

Response: Thanks for the positive feedback. We align the words in Tab. 1 to the left as many previous published articles do.

Reviewer #3 (Remarks to the Author):

The authors have addressed my comments from the previous round of review, and I recommend publication.

Response: Thanks for the recommendation of publication.